# Endotracheal tube, by the venturi effect, reduces the efficacy of increasing inlet pressure in improving pendelluft

**Kazuhiro Takahashi**[1]*, **Hiroaki Toyama**[1], **Yutaka Ejima**[2], **Jinyou Yang**[3], **Kenji Kikuchi**[4], **Takuji Ishikawa**[5], **Masanori Yamauchi**[1]

**1** Anesthesiology and Perioperative Medicine, Tohoku University Graduate School of Medicine, Sendai, Japan, **2** Division of Surgical Center and Supply, Sterilization, Tohoku University Hospital, Sendai, Japan, **3** Department of Biophysics, School of Intelligent Medicine, China Medical University, Shenyang, China, **4** Department of Finemechanics, Graduate School of Engineering, Tohoku University, Sendai, Japan, **5** Graduate School of Biomedical Engineering, Tohoku University, Sendai, Japan

* takahashi.kazuhiro@med.tohoku.ac.jp

**Data Availability Statement:** All relevant data are within the paper.

**Funding:** The authors received no specific funding for this work.

## Abstract

In mechanically ventilated severe acute respiratory distress syndrome patients, spontaneous inspiratory effort generates more negative pressure in the dorsal lung than in the ventral lung. The airflow caused by this pressure difference is called pendelluft, which is a possible mechanisms of patient self-inflicted lung injury. This study aimed to use computer simulation to understand how the endotracheal tube and insufficient ventilatory support contribute to pendelluft. We established two models. In the invasive model, an endotracheal tube was connected to the tracheobronchial tree with 34 outlets grouped into six locations: the right and left upper, lower, and middle lobes. In the non-invasive model, the upper airway, including the glottis, was connected to the tracheobronchial tree. To recreate the inspiratory effort of acute respiratory distress syndrome patients, the lower lobe pressure was set at -13 $cmH_2O$, while the upper and middle lobe pressure was set at -6.4 $cmH_2O$. The inlet pressure was set from 10 to 30 $cmH_2O$ to recreate ventilatory support. Using the finite volume method, the total flow rates through each model and toward each lobe were calculated. The invasive model had half the total flow rate of the non-invasive model (1.92 L/s versus 3.73 L/s under 10 $cmH_2O$, respectively). More pendelluft (gas flow into the model from the outlets) was observed in the invasive model than in the non-invasive model. The inlet pressure increase from 10 to 30 $cmH_2O$ decreased pendelluft by 11% and 29% in the invasive and non-invasive models, respectively. In the invasive model, a faster jet flowed from the tip of the endotracheal tube toward the lower lobes, consequently entraining gas from the upper and middle lobes. Increasing ventilatory support intensifies the jet from the endotracheal tube, causing a venturi effect at the bifurcation in the tracheobronchial tree. Clinically acceptable ventilatory support cannot completely prevent pendelluft.

**Competing interests:** The authors have declared that no competing interests exist.

## Introduction

Patients with severe acute respiratory distress syndrome (ARDS) need mechanical ventilation for life support. Spontaneous breathing has long been considered desirable during mechanical ventilation because it maintains contraction of the dorsal diaphragm, resulting in improved gas exchange and air content distribution. However, recent studies have demonstrated that strong inspiratory effort aggravates lung injury in the aforementioned patients through a mechanism called patient self-inflicted lung injury (P-SILI) [1–3].

Gas movement in a patients' lung, called pendelluft, is considered one of the mechanisms of P-SILI [4]. In a healthy lung, the negative pressure generated by the diaphragm is propagated across the lung surface equally (fluid-like behavior). In contrast, in patients with severe ARDS, lung injury is heterogeneous, so the dorsal lung's mechanics differ from that of the ventral lung [5, 6]. Hence, an injured lung localizes the negative pressure to the dorsal lung (solid-like behavior). When the air supply from the ventilator is insufficient relative to the patient's inspiratory effort, gas movement is directed from the ventral to the dorsal lung (in most cases, from the upper to the lower lobe) in the early inspiratory phase [7]. This gas movement, the pendelluft, is accompanied by hyperextension and severe inflammation in the lower lobe [8, 9]. Contrastingly, the upper lobe is initially entrained and subsequently inflated during the inspiratory phase because the compliance of the upper lobe is greater than that of the lower lobe [10]. This repeated collapse and re-inflation cycle in the upper lobe is considered harmful [11, 12]. Additionally, the gas movement caused by the pendelluft will not contribute to gaseous exchange, thus worsening the efficacy of ventilation and increasing the work of breathing (WOB) [7].

Recently, the ventilatory pressure and endotracheal tube (ETT) resistance were reported to affect the air supply from a ventilator toward the lungs, thus affecting pendelluft [7, 13]. Regarding ventilatory pressure, a recent clinical study found that sufficient ventilatory pressure support is required to prevent pendelluft for patients with severe ARDS and strong inspiratory effort [7]. Inadequate ventilatory pressure support may exacerbate lung injury [11, 14]. Contrarily, ETT resistance is reported as the leading cause of lung edema [15, 16] and pendelluft [13] under spontaneous breathing conditions. The pressure dissipation by the ETT affects the pressure delivered to the patient's airway and gas flow in the patient's lung because the pressure delivered to the airways is a residue of ventilator-generated pressure minus pressure dissipation caused by the ETT [17–20].

Pendelluft can be detected with dynamic computed tomography or electrical impedance tomography (EIT) in a clinical setting [11]. However, these tests cannot evaluate the gas flow in the airway tree. Regarding this aspect, a computer simulation study is useful because it can calculate and visualize the gas flow in the airway tree, dealing with morphological factors such as the branches' angles and changes in cross-sectional areas [21–24]. Recently, some computational gas flow analyses simulating the connection between a patient's airway and an ETT were performed. These studies showed that the gas flow in the patient's airway was affected by the faster jet flow through the ETT [25–30].

To study the gas flow associated with pendelluft, a computer simulation study is suitable because it can handle the inter-lobar pressure differences caused by the solid-like behavior. However, computer simulations comparing an intubated patient model with a non-intubated patient model under various ventilatory conditions have not been conducted. By comparing these two models, we can examine the ETT's role as an external resistance and jet flow distributer and how the ventilatory support affects it. Hence, using computer simulations, in this study we investigated how ETT and ventilatory support contribute to the pendelluft phenomenon in patients with severe ARDS.

## Materials and methods

### Simulation modeling

We created a non-intubated simulation model named the non-invasive model (Fig 1A) and an intubated simulation model named the invasive model (Fig 1B) using ANSYS SpaceClaim (ANSYS Inc., Canonsburg, PA, USA).

The invasive model comprised ETT and tracheobronchial tree parts. The ETT part included a sequence of a slip joint (inner diameter [ID] 11.6 mm, length 1.5 cm) plus a tube element (ID 8 mm, length 30 cm). Furthermore, the model of the tracheobronchial tree was patterned based on the work by Kitaoka et al. [31, 32] with the trachea (a regular 16-cornered square, 3.72 mm long on each side, and 3 cm in length) and tracheal bifurcation sections and subsequent 34 outlets.

Meanwhile, the non-invasive model comprised upper airway and tracheobronchial tree portions. The upper airway followed the sequence of the oral cavity, glottis, and extended trachea. The oral cavity was modeled as a cylinder (ID 19 mm, length 13.5 cm). The shape of the oral cavity had no specific geometry because the purpose of this study did not include an analysis of intraoral airflow. The glottis was modeled as a cylinder (ID 10 mm, length 0.5 cm). The extended trachea was modeled as the regular 16-cornered square, 3.72 mm long on each side and 10 cm in length. In summary, the total upper airway length was 24 cm, corresponding to the length at which an ETT (ID 8 mm) is often fixed at the corner of the mouth. This upper airway model was connected to the above-mentioned tracheobronchial tree part.

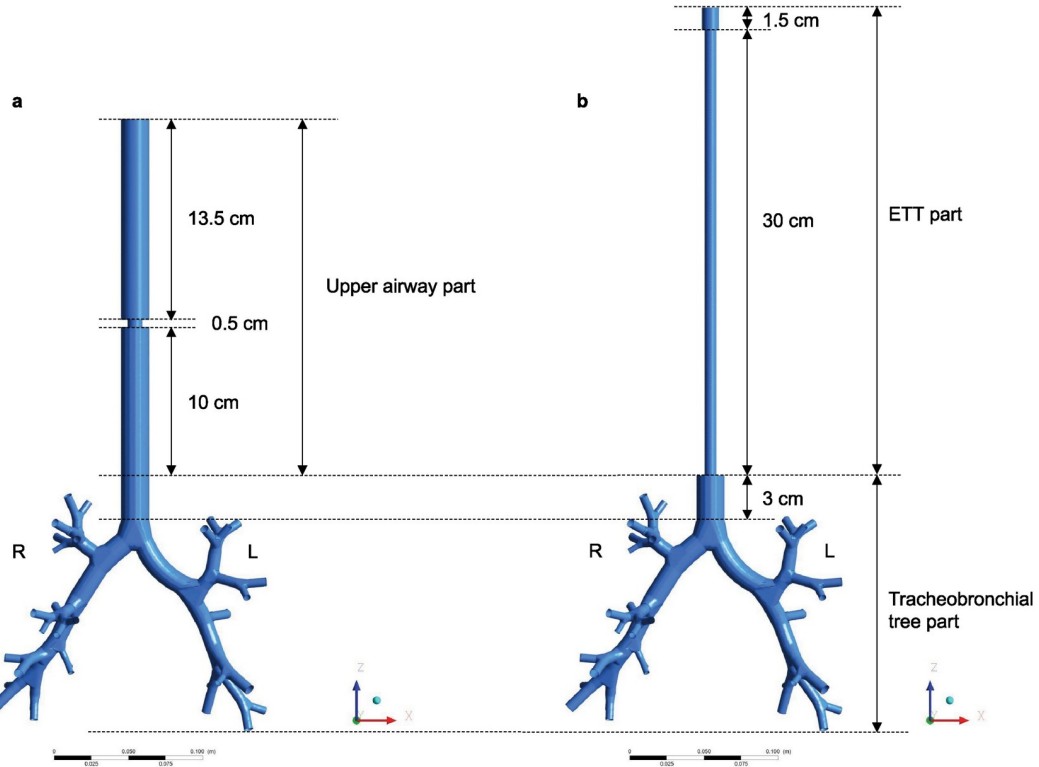

**Fig 1. Simulation models.** (A) The non-invasive model in which the upper airway, including the glottis, is connected to the tracheobronchial tree. (B) The invasive model in which the endotracheal tube is connected to the tracheobronchial tree. R: Right; L: Left; ETT: Endotracheal tube.

After generating the two models, ANSYS ICEM (Ansys Inc.) was used to create the mesh. The computational mesh was generated using unstructured tetrahedral elements with two layers of prism mesh, considered appropriate for our complex lung geometry. In line with previous studies, we adopted mesh numbers of $8.51*10^5$ and $1.06*10^6$ for the intubated-tree and glottis-tree models, respectively [25].

## Simulation details

ANSYS CFX (ANSYS Inc., Canonsburg, PA, USA) was used for steady-state analysis. Continuity and Navier–Stokes equations were solved under isothermal conditions using a finite volume method. K-ε was used as the turbulence model, using a medium turbulence intensity. The convergence criteria were set such that the root-mean-square residuals for mass and momentum equations were less than $10^{-4}$. The inlet boundary condition was provided as the total pressure to simulate the initial inspiratory flow during pressure-controlled ventilation. The 34 outlets of the two models were divided into six groups, namely the right upper (RU), right middle (RM), right lower (RL), left upper (LU), left middle (LM), and left lower (LL) outlets. The outlet boundary conditions were set as static pressure to match each simulation described below. The models' walls were smooth, and no-slip wall boundary conditions were used. The gas density used for the computer simulation was 1.26 kg/m$^3$, assuming 37˚C oxygen.

## Preliminary study: Verification of simulation accuracy

Previous experimental studies have reported the pressure drop across an ETT with ID 8.0 mm [17], proving that this drop depends on the flow rate. Hence, in this simulation, we examined the reliability of our simulation models and methods using the ETT created for the invasive model. Here, the static pressure difference between the ETT's inlet (slip joint side) and outlet was defined as the pressure drop. We set the inlet velocity from 1 to 30 m/s (inlet flow rate 0.1 to 3.2 L/s); under each condition, a pressure drop across the ETT was simulated.

## Main study: Flow delivery to the six outlet groups under various boundary conditions

In this part of the study, the gas flow toward each of the six outlet groups was computed for the non-invasive and invasive models under various conditions. In a computer simulation study, conditions given to the inlet and outlet of the model are called boundary conditions. In this study, boundary conditions of the inlet and outlet were set as pressure. Hence, ventilatory support from the ventilator was expressed as positive inlet pressure. Therefore, to simulate stronger ventilatory support, we applied a higher inlet pressure for the models. Contrarily, a patient's inspiratory effort was expressed as negative outlet pressure. Therefore, to simulate stronger inspiratory effort, we applied more negative outlet pressure for the models. For the two models, we applied three levels of inlet pressure (namely high, middle, and low support) and seven outlet pressure conditions (namely non-effort, [strong, moderate, and weak] solid-like effort, and [strong, moderate, and weak] fluid-like effort). The seven outlet boundary conditions are shown in Fig 2. Details of these boundary conditions are described below. Then, the flow rate through each outlet group was simulated using ANSYS CFX.

Under the boundary conditions, the gas flow out of the model through the outlets meant lung lobe inflation, whereas the gas flow into the model through the outlets meant lung lobe deflation. Therefore, in this study, gas flow into the model through the outlets was defined as the pendelluft.

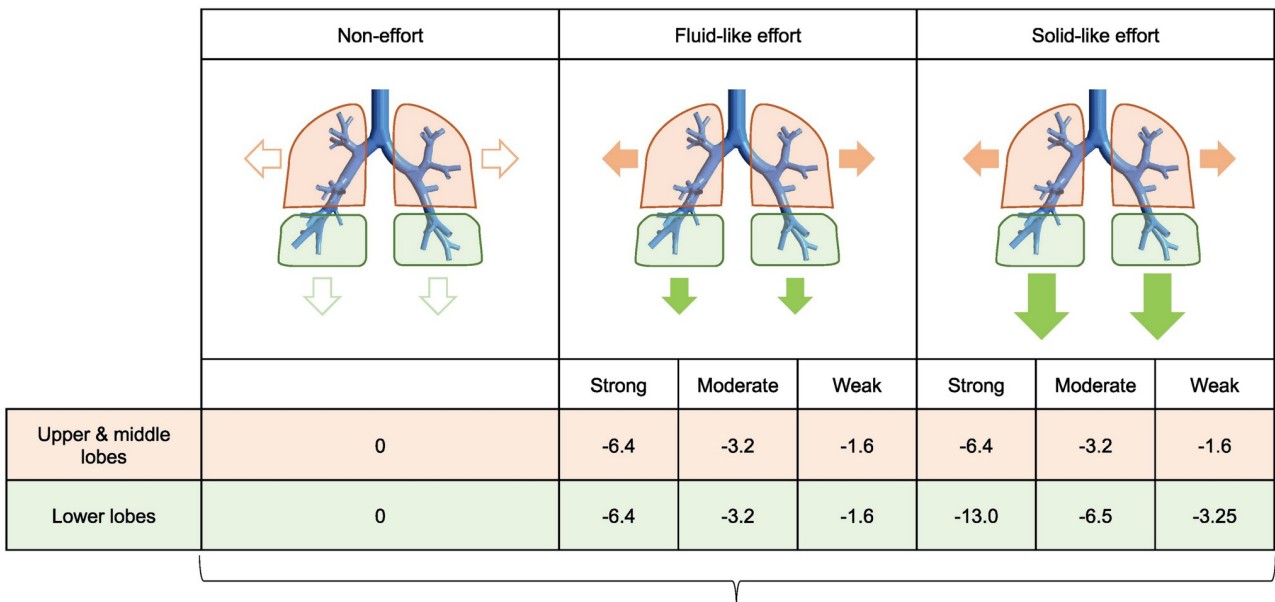

| | Non-effort | Fluid-like effort | | | Solid-like effort | | |
|---|---|---|---|---|---|---|---|
| | | Strong | Moderate | Weak | Strong | Moderate | Weak |
| Upper & middle lobes | 0 | -6.4 | -3.2 | -1.6 | -6.4 | -3.2 | -1.6 |
| Lower lobes | 0 | -6.4 | -3.2 | -1.6 | -13.0 | -6.5 | -3.25 |

Seven Outlet boundary conditions

**Fig 2. The outlet boundary conditions in the main study.** The left, center, and right pattern diagrams indicate the non-effort, fluid-like effort, and solid-like effort conditions, respectively. The size of the vector represents the strength of the effort. The fluid-like effort and solid-like effort conditions have three degrees of strength named strong, moderate, and weak. Each number below the pattern diagram indicates the pressure applied at each outlet [$cmH_2O$].

### Details of boundary conditions

Regarding the inlet boundary conditions, we set a total pressure of 10 $cmH_2O$ for a low level of ventilatory support. Similarly, we set a pressure of 20 $cmH_2O$ for a middle level and that of 30 $cmH_2O$ for a high level of ventilatory support.

Regarding the outlet boundary conditions as shown in Fig 2, we set a static pressure of 0 $cmH_2O$ at all of the six outlet groups (upper, middle, and lower lobes) for the non-effort condition. For the strong solid-like effort condition, we applied a negative pressure of -13.0 $cmH_2O$ at the lower lobes and -6.4 $cmH_2O$ at the upper and middle lobes, assuming that patients with ARDS exert strong solid-like efforts, as reported previously [12]. For the moderate solid-like effort condition, we applied half of the negative pressure for the strong condition, that is, -6.5 $cmH_2O$ at the lower lobes and -3.2 $cmH_2O$ at the upper and middle lobes. For the weak solid-like effort condition, we applied half of the negative pressure for the moderate condition, that is, -3.25 $cmH_2O$ at the lower lobes and -1.6 $cmH_2O$ at the upper and middle lobes. For the strong, moderate, and weak fluid-like effort conditions, we applied a negative pressure of -6.4, -3.2, and -1.6 $cmH_2O$ at the upper, middle, and lower lobes, respectively, to facilitate comparison with the solid-like effort conditions. Then, we conducted 42 types of computer simulations (2 models × 3 inlet conditions × 7 outlet conditions) and recorded the gas flow through the inlet and each of the six outlet groups. Further, using some of the computer simulation data obtained in the main study, we conducted two comparative research described below.

**Comparative research 1: Effect of ETT.**   The purpose of this comparative research was to investigate the effect of an ETT on pendelluft, using invasive and non-invasive models. The high ventilatory support condition at the inlet and the three types of inspiratory (non-effort, strong fluid-like effort, and strong solid-like-effort) conditions at the outlets were applied, and

the gas flow rate through the two models was compared. Additionally, gas flow vectors in the two models under high inlet pressure and non-effort condition were evaluated.

**Comparative research 2: Effect of ventilatory support.** The purpose of this comparative research was to investigate the effect of the ventilatory support level on pendelluft. We used invasive and non-invasive models. (A) First, the three levels (high, middle, and low) of ventilatory support conditions at the inlet and strong solid-like-effort condition at the outlets were applied, and the gas flow rate through the six outlet groups were compared. (B) Second, the three levels (high, middle, and low) of ventilatory support conditions at the inlet and non-effort condition at the outlets were applied, and the gas flow rate through the RU outlet was compared among the levels.

## Results

Fig 3 shows the result of the preliminary study. The gray line shows the pressure-drop curve based on the previous experimental report by Flevari et al. [17], and the black line shows the pressure-drop curve of our ETT model. The vertical axis indicates the pressure drop [cmH$_2$O] across the ETT, while the horizontal axis indicates the flow rate through the ETT [L/s]. In line with that in the aforementioned report, the pressure-drop curve obtained using our computer simulation showed that the pressure drop is a quadratic function of the flow rate. This verification study suggests that our simulation calculations were close to the experimental values under general gas flow rate conditions.

Table 1 shows the gas flow rate [L/s] through the inlet and each of the six outlet groups for the 42 types of computer simulations (2 models × 3 inlet levels × 7 outlet conditions). Table 1A shows the non-invasive model, and Table 1B shows the invasive model. The gas flow into the model was positive, and the gas flow out of the model was negative.

### Comparative research 1: Effect of ETT

Fig 4 shows the results of comparative research 1 of the main study. The gas flow rate through the inlet and each of the six outlet groups are depicted. Fig 4A–4C indicate the non-effort, strong fluid-like effort, and strong solid-like effort conditions, respectively. The gray and black

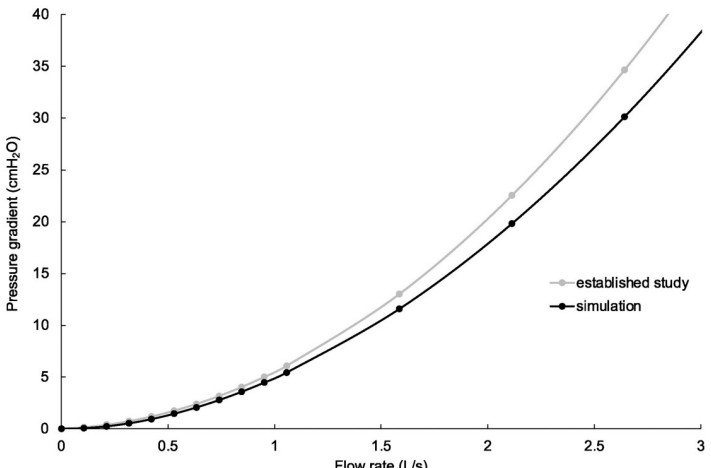

**Fig 3. The relationship between the pressure drop across and flow rate through the endotracheal tube.** The gray line shows the pressure drop curve based on previous studies. The black dots show the pressure drop obtained in our simulation, and the black line shows the approximate curve. The simulation values were close to the values of established studies under general flow rate conditions.

**Table 1.** **A.** Non-invasive Model. **B.** Invasive Model.

| Boundary conditions | | | Gas flow rate [L/s] | | | | | | |
|---|---|---|---|---|---|---|---|---|---|
| **A)** | | | | | | | | | |
| Inlet | Outlet | Intensity | inlet | LL | LM | LU | RL | RM | RU |
| low | non | | 2.85 | -0.79 | -0.21 | -0.29 | -0.87 | -0.32 | -0.37 |
| middle | non | | 4.00 | -1.15 | -0.30 | -0.38 | -1.27 | -0.45 | -0.45 |
| high | non | | 4.88 | -1.42 | -0.37 | -0.44 | -1.58 | -0.55 | -0.52 |
| low | fluid | strong | 3.62 | -1.03 | -0.27 | -0.35 | -1.14 | -0.41 | -0.43 |
| low | fluid | moderate | 3.26 | -0.91 | -0.24 | -0.33 | -1.01 | -0.37 | -0.41 |
| low | fluid | weak | 3.07 | -0.85 | -0.23 | -0.31 | -0.94 | -0.35 | -0.39 |
| middle | fluid | strong | 4.57 | -1.33 | -0.34 | -0.42 | -1.47 | -0.52 | -0.50 |
| middle | fluid | moderate | 4.30 | -1.24 | -0.32 | -0.40 | -1.37 | -0.49 | -0.48 |
| middle | fluid | weak | 4.16 | -1.20 | -0.31 | -0.39 | -1.33 | -0.47 | -0.46 |
| high | fluid | strong | 5.35 | -1.57 | -0.40 | -0.48 | -1.74 | -0.61 | -0.56 |
| high | fluid | moderate | 5.12 | -1.50 | -0.38 | -0.46 | -1.66 | -0.58 | -0.54 |
| high | fluid | weak | 5.01 | -1.46 | -0.38 | -0.45 | -1.62 | -0.57 | -0.53 |
| low | solid | strong | 3.73 | -3.45 | 0.47 | 0.81 | -3.54 | 0.99 | 0.99 |
| low | solid | moderate | 3.30 | -2.58 | 0.28 | 0.47 | -2.65 | 0.63 | 0.55 |
| low | solid | weak | 3.07 | -1.94 | 0.14 | 0.22 | -2.09 | 0.39 | 0.21 |
| middle | solid | strong | 4.63 | -3.63 | 0.39 | 0.67 | -3.73 | 0.89 | 0.78 |
| middle | solid | moderate | 4.30 | -2.75 | 0.20 | 0.32 | -2.94 | 0.56 | 0.30 |
| middle | solid | weak | 4.15 | -2.11 | 0.02 | 0.04 | -2.33 | 0.23 | 0.01 |
| high | solid | strong | 5.38 | -3.78 | 0.33 | 0.56 | -3.90 | 0.80 | 0.61 |
| high | solid | moderate | 5.12 | -2.88 | 0.12 | 0.20 | -3.13 | 0.44 | 0.14 |
| high | solid | weak | 5.01 | -2.27 | -0.10 | -0.10 | -2.52 | 0.03 | -0.05 |
| **B)** | | | | | | | | | |
| Inlet | Outlet | Intensity | Inlet | LL | LM | LU | RL | RM | RU |
| low | non | | 1.38 | -0.55 | -0.11 | -0.16 | -0.43 | -0.14 | -0.01 |
| middle | non | | 1.99 | -0.82 | -0.15 | -0.22 | -0.62 | -0.19 | 0.01 |
| high | non | | 2.47 | -1.03 | -0.19 | -0.27 | -0.77 | -0.24 | 0.03 |
| low | fluid | strong | 1.79 | -0.72 | -0.14 | -0.20 | -0.56 | -0.17 | 0.00 |
| low | fluid | moderate | 1.60 | -0.64 | -0.12 | -0.18 | -0.50 | -0.16 | 0.00 |
| low | fluid | weak | 1.49 | -0.59 | -0.11 | -0.17 | -0.46 | -0.15 | -0.01 |
| middle | fluid | strong | 2.30 | -0.96 | -0.18 | -0.25 | -0.72 | -0.22 | 0.03 |
| middle | fluid | moderate | 2.15 | -0.89 | -0.17 | -0.24 | -0.67 | -0.21 | 0.02 |
| middle | fluid | weak | 2.08 | -0.85 | -0.16 | -0.23 | -0.64 | -0.20 | 0.01 |
| high | fluid | strong | 2.72 | -1.14 | -0.21 | -0.29 | -0.86 | -0.26 | 0.04 |
| high | fluid | moderate | 2.60 | -1.09 | -0.20 | -0.28 | -0.82 | -0.25 | 0.04 |
| high | fluid | weak | 2.53 | -1.05 | -0.19 | -0.28 | -0.79 | -0.24 | 0.02 |
| low | solid | strong | 1.92 | -3.10 | 0.63 | 1.10 | -3.08 | 1.10 | 1.43 |
| low | solid | moderate | 1.66 | -2.27 | 0.41 | 0.71 | -2.25 | 0.77 | 0.96 |
| low | solid | weak | 1.51 | -1.70 | 0.26 | 0.44 | -1.66 | 0.54 | 0.62 |
| middle | solid | strong | 2.38 | -3.21 | 0.58 | 1.01 | -3.19 | 1.08 | 1.35 |
| middle | solid | moderate | 2.18 | -2.41 | 0.36 | 0.61 | -2.34 | 0.75 | 0.86 |
| middle | solid | weak | 2.07 | -1.82 | 0.19 | 0.31 | -1.78 | 0.51 | 0.52 |
| high | solid | strong | 2.78 | -3.32 | 0.53 | 0.93 | -3.25 | 1.06 | 1.27 |
| high | solid | moderate | 2.61 | -2.53 | 0.31 | 0.53 | -2.43 | 0.72 | 0.80 |
| high | solid | weak | 2.53 | -1.92 | 0.13 | 0.21 | -1.86 | 0.46 | 0.44 |

LL: left lower, LM: left middle, LU: left upper, RL: right lower, RM: right middle, RU: right upper.

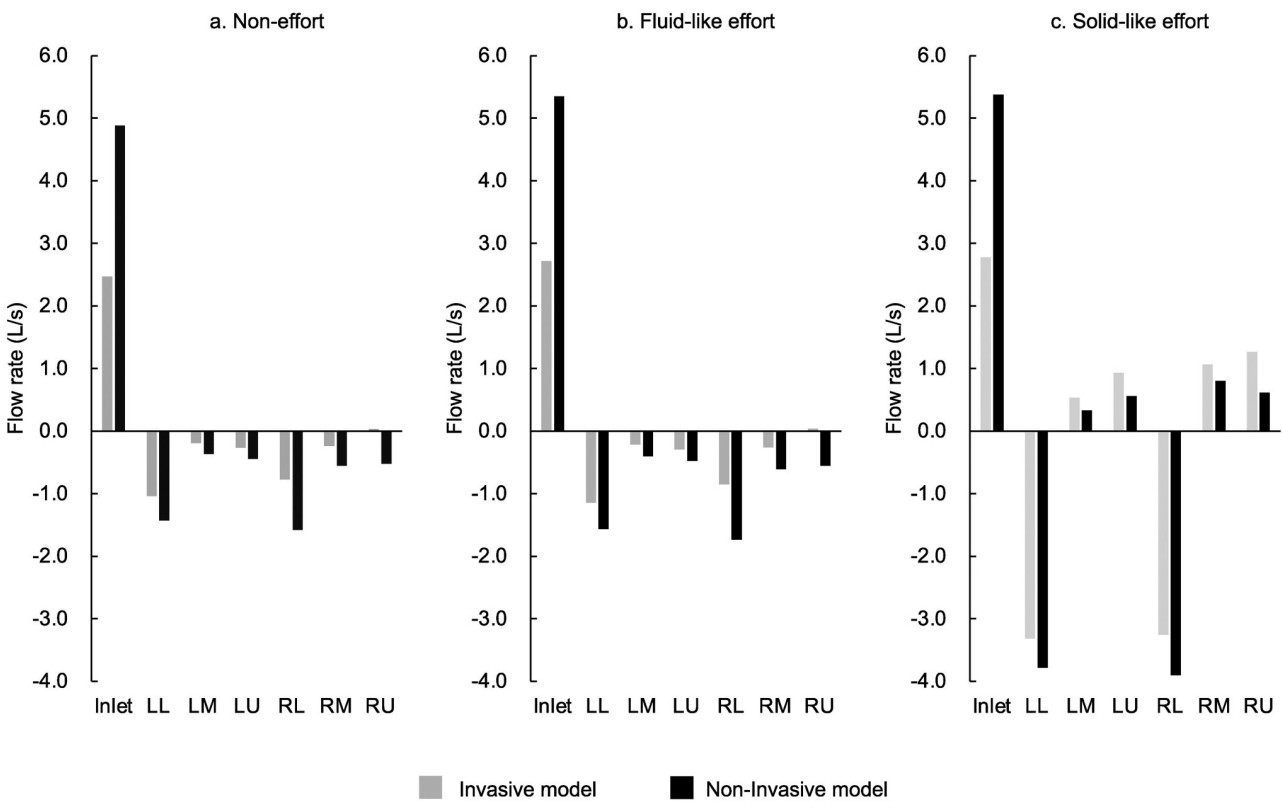

**Fig 4. Gas flow through the invasive and non-invasive models.** Gas flow rates through the inlet and each of the six outlet groups in the invasive and non-invasive models, under the non-effort (A), strong fluid-like effort (B), and strong solid-like effort (C) conditions. The gray and black bar graphs in each figure indicate the invasive model and non-invasive models, respectively. The gas flow into the models was described as positive, and the gas flow out of the models was described as negative. LL: Left lower; LM: Left middle; LU: Left upper; RL: Right lower; RM: Right middle; RU: Right upper.

bar graphs in each figure indicate the invasive and non-invasive models, respectively. The gas flow into the models was described as positive, and the gas flow out of the models was described as negative. Therefore, the positive value observed at outlets is defined as the pendel-luft. The non-effort (Fig 4A) and strong fluid-like effort (Fig 4B) conditions show similar gas flow distribution patterns. Conversely, the strong solid-like effort (Fig 4C) condition indicates that the gas flow out of the lower lobes (LL + RL) increased, and the pendelluft was observed at the other outlets. Comparing the two models, the inlet gas flow in the invasive model was less than that in the non-invasive model. Moreover, the gas flow out of the lower lobes was enhanced in the invasive model. Specifically, in the non-invasive model, the gas flow out of the lower lobes was equivalent to 61%, 62%, and 143% of the inlet gas flow under the non-effort, fluid-like effort, and solid-like effort conditions, respectively. In the invasive model, the gas flow out of the lower lobes was equivalent to 73%, 74%, and 236% of the inlet gas flow under the non-effort, fluid-like effort, and solid-like effort conditions, respectively.

Fig 5 indicates the gas flow vectors inside the two models under high inlet pressure and the non-effort condition (Fig 4A condition) in comparative research 1. The vector length indicates the velocity of the gas flow at each point, and the background colors indicate the energy (total pressure) of the gas on the Y = 0 cross-sectional area in both the invasive and non-invasive models. Red color indicates high energy, and blue indicates low energy. In the invasive model, a faster jet from the ETT flowed along the central axis in the tracheobronchial tree toward the

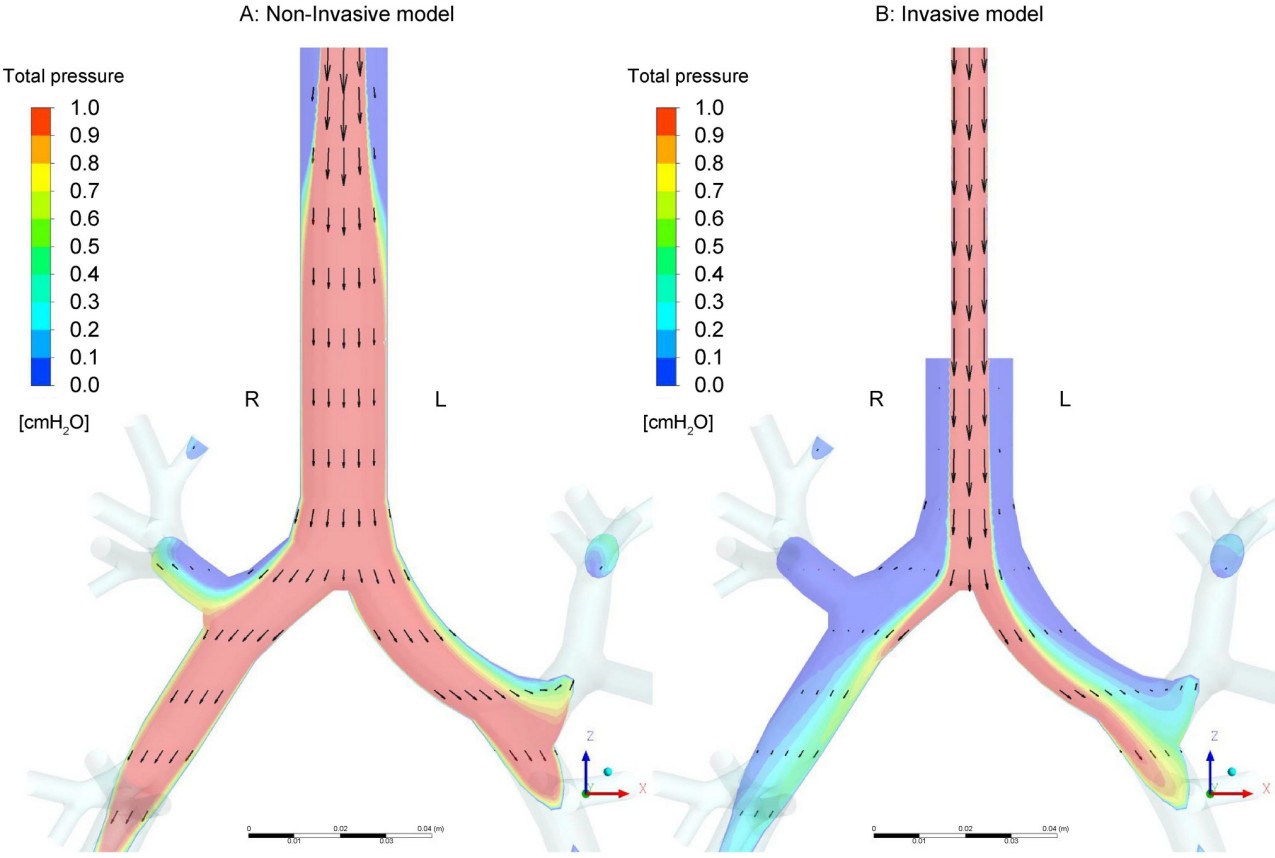

**Fig 5. Vectors inside the two simulation models.** Y = 0 depicts vectors on the cross-sections under high inlet pressure and the non-effort condition. Fig 5A indicates the non-invasive model, while Fig 5B shows the invasive model. The length of the vector indicates the velocity of the gas flow at each point, and the background colors indicate the energy (total pressure) of the gas on the Y = 0 cross-sectional area. The distribution of the energy is more heterogeneous in the invasive model than in the non-invasive model.

lower lobes, resulting in higher velocity and energy. Accordingly, the energy distribution in the invasive model was more heterogeneous than that in the non-invasive model.

Fig 6 shows the detailed gas flow in the invasive model under high inlet pressure and the non-effort condition (Fig 4A condition) in comparative research 1. The arrowheads and background colors indicate the direction and velocity of the gas flow at each point, respectively. A faster jet from the ETT flowed along the central axis in the tracheobronchial tree toward the lower lobes. Further, entrained flow from the RU outlet joining the faster jet toward RL was visualized (venturi effect).

## Comparative research 2: Effect of ventilatory support

Fig 7 shows the results of comparative research 2 (A) of the main study. Fig 7A and 7B show the invasive and non-invasive models, respectively, indicating gas flow rate through the inlet and each of the six outlet groups under the strong solid-like effort condition. The bar graph with a thin horizontal line, bar graph with a bold horizontal line, and filled bar graph in each figure indicate the low, middle, and high inlet ventilatory support conditions, respectively. The gas flow into the models was described as positive, and the gas flow out of the models was described as negative. The total gas flow into the model from the outlets under each condition was defined as the pendelluft in this study. In the invasive model under the low support

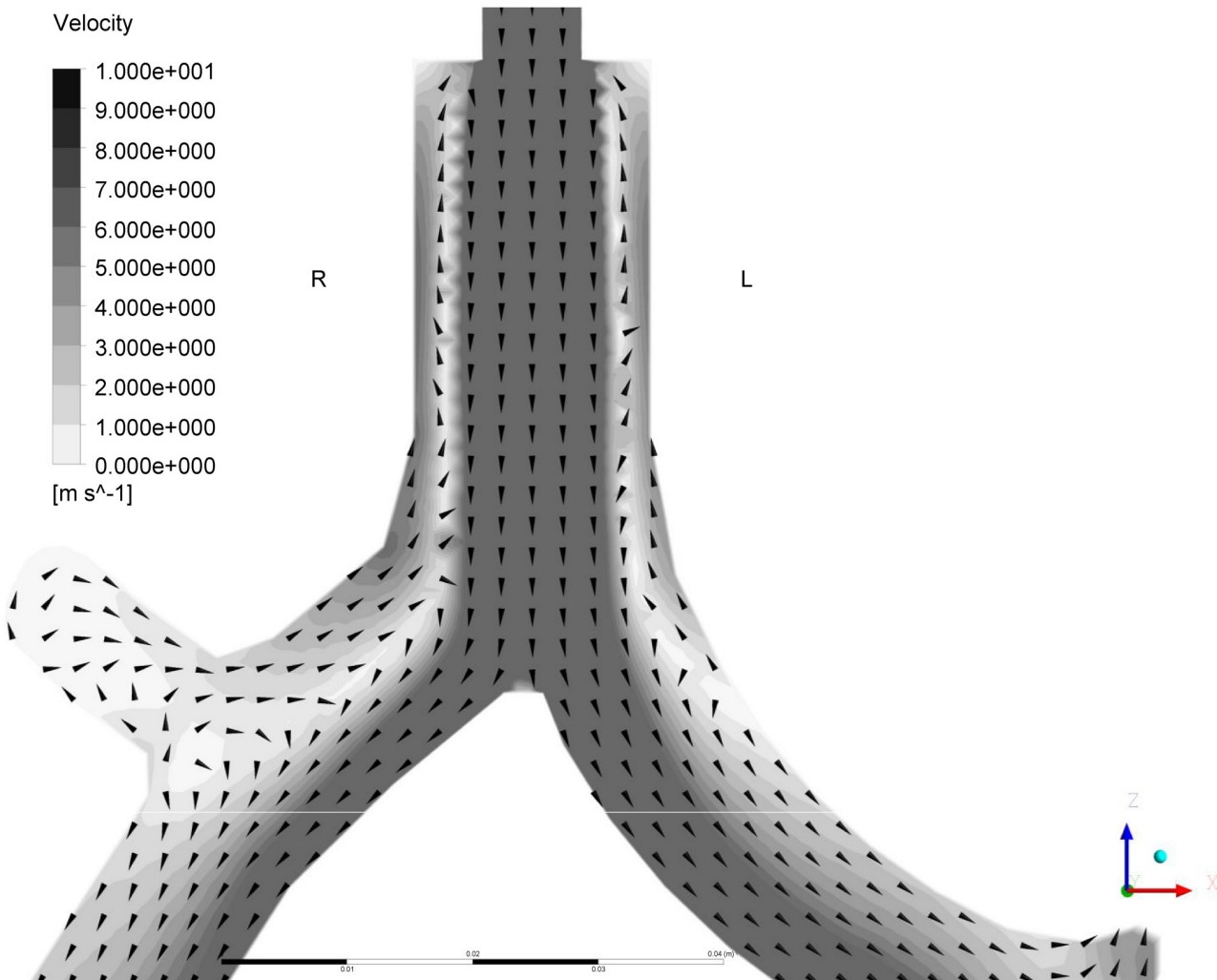

**Fig 6. Detailed gas flow in the invasive model.** Y = 0 depicts detailed gas flow on the cross-sections under high inlet support and the non-effort condition in the invasive model. The arrowheads and background colors indicate the direction and velocity of the gas flow at each point, respectively. R: Right; L: Left.

condition, the pendelluft was calculated as 4.26 L/s (1.43 from the RU lobe + 1.10 from the RM lobe + 1.10 from the LU lobe + 0.63 from the LM lobe). Under the middle and high support conditions, the pendelluft was calculated as 4.01 L/s and 3.79 L/s, respectively (6% and 11% decrease from the low support condition, respectively). In the non-invasive model, an increase in the inlet ventilatory support condition from low to middle and high support decreased the pendelluft by 16% and 29%, respectively.

Fig 8 shows the result of comparative research 2 (B) of the main study. The gray bar graph with a thin vertical line, gray bar graph with a bold vertical line, and gray filled bar graph indicate low, middle, and high inlet ventilatory pressure, respectively, in the invasive model. The black bar graph with a thin vertical line, black bar graph with a bold vertical line, and black filled bar graph indicate low, middle, and high inlet ventilatory pressure, respectively, in the non-invasive model. The gas flow into the RU was described as positive, and the gas flow out of the RU was described as negative.

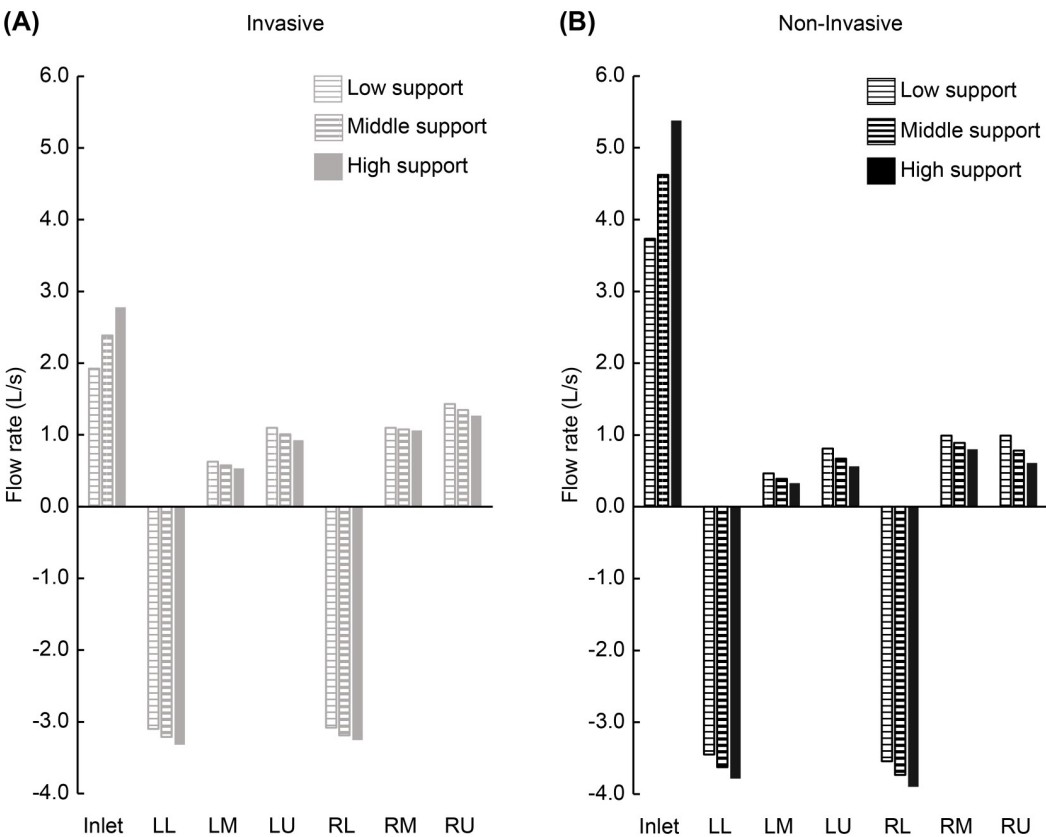

**Fig 7. Effect of ventilatory support on pendelluft.** Gas flow rates through the inlet and each of the six outlet groups under the strong solid-like effort condition in the invasive (A) and non-invasive models (B) are depicted. The bar graph with a thin horizontal line, bar graph with a bold horizontal line, and filled bar graph in each figure indicate the low, middle, and high inlet ventilatory support conditions, respectively. The gas flow into the models was described as positive, and the gas flow out of the models was described as negative. LL: Left lower; LM: Left middle; LU: Left upper; RL: Right lower; RM: Right middle; RU: Right upper.

## Discussion

Pendelluft is the air movement within a lung caused by multiple reasons, including the heterogeneity of the lung mechanics, intensity of mechanical ventilatory support, and inspiratory effort [7, 33, 34]. In a clinical setting, pendelluft is traditionally identified and investigated using a rapid airway occlusion technique at the end of inspiration for deeply sedated and paralyzed patients [33]. The pendelluft during the end-inspiratory phase is caused by the time constant's heterogeneity and lung's viscoelasticity. As Alessandro et al. [34] reported, the ventral lung appears to have characteristics that allow it to be more inflated than the dorsal lung under high potential energy. Conversely, with the development of EIT, pendelluft occurring in the early-inspiratory phase can be detected. EIT can be used for patients who maintain spontaneous breathing. The solid-like behavior of the intra-lung pressure distribution under inspiratory effort and many ventilatory and patient-related factors cause this type of pendelluft.

Recently, pendelluft has been considered one of the mechanisms of P-SILI, a concept wherein patients' inspiratory effort causes regional edema, tidal recruitment, and distortion [1, 2, 35, 36]. The inhomogeneous gas distribution of pendelluft is assumed to increase the strain and amplify the stress at the interface between open and closed pulmonary units (= stress raiser), thereby damaging the lung [1, 37–40]. To date, pendelluft is reported to be associated

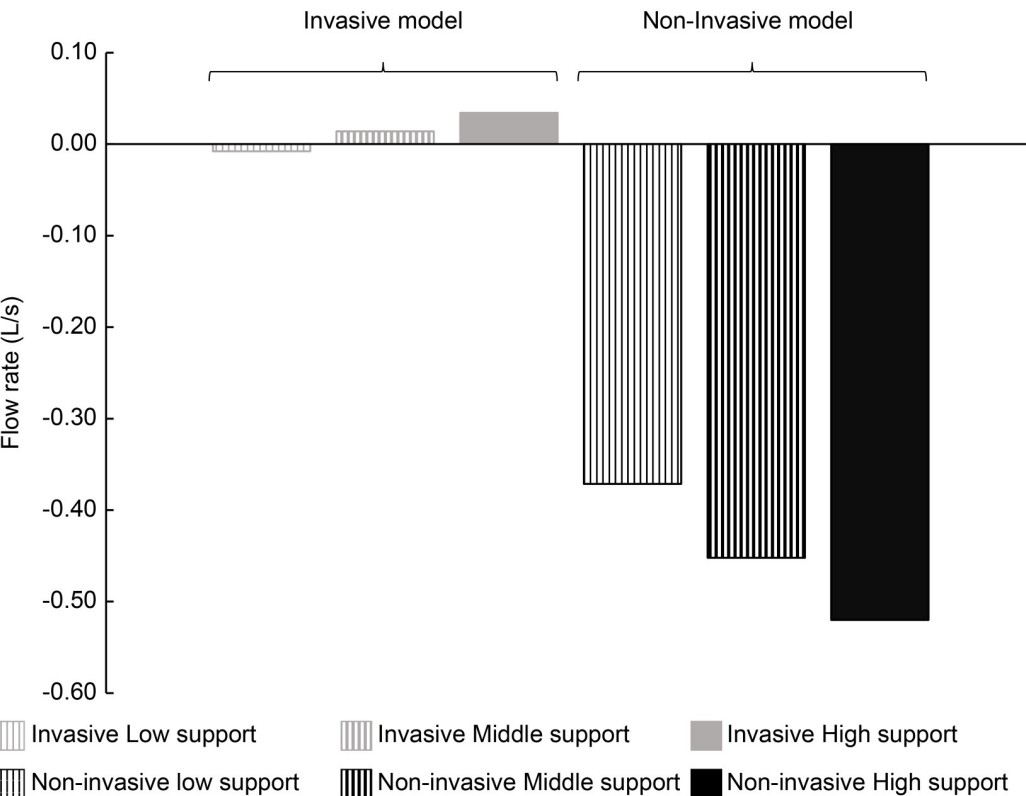

**Fig 8. Effect of ventilatory support on the RU outlet.** The gas flow rate through the RU outlet under the non-effort condition in the invasive and non-invasive models are depicted. The gray bar graph with a thin vertical line, bar graph with a bold vertical line, and filled bar graph indicate the low, middle, and high inlet support conditions, respectively, in the invasive model. The black bar graph with a thin vertical line, bar graph with a bold vertical line, and filled bar graph indicate low, middle, and high inlet ventilatory pressure, respectively, in the non-invasive model.

with longer ventilation duration in patients with severe ARDS [41], whereas its effect on patients with mild ARDS is unclear.

Despite several studies having focused on the harmful effects of pendelluft, its leading causative factors are not fully understood. To investigate these factors, a computer simulation study is suitable because it can handle a three-dimensional model and visualize the airflow inside the tracheobronchial tree under arbitrary boundary conditions imitating various respiratory status, such as the intensity of mechanical support and inspiratory effort [25–30].

In this study, we used computer simulation to investigate how an ETT and ventilatory support affect pendelluft during the early phase of inspiration. In the preliminary study, we examined the reliability of our simulation system. An acceptable error was observed while comparing the pressure-drop data obtained using our simulation to the experimental report [17]. Further, using the system, we conducted the main study to investigate the pendelluft under several patterns of the boundary conditions. The results of the main study were consistent with clinical knowledge and the fluid mechanics theory. First, as shown in Table 1A (non-invasive model) and 1B (invasive model), the inlet gas flow rate increases as inlet pressure increases, under the same outlet boundary conditions. Second, when comparing fluid-like and solid-like effort patterns under the same inlet pressure, the inlet gas flow rates are almost the same, although the outlet gas flow patterns are very different. Specifically, under solid-like effort, gas flows out of the lower lobes and flows into the model from the other lobes. This gas

flow from the non-dependent toward the dependent lung without changing inlet gas flow represents the characteristics of pendelluft. As expected, pendelluft increased with a stronger inspiratory effort (Table 1). Third, more flow was directed toward the lower lobe, even under non-effort or fluid-like effort conditions. This result is consistent with a previous report showing that the volume change from expiration to inspiration was greater in the lower lobe than in the upper lobe among healthy volunteers [42]. In this study, besides evaluating this clinical knowledge, we focused on the following two comparative studies to understand pendelluft more morphologically.

## Comparative research 1: Effect of ETT

Here, we discuss the effect of an ETT on the gas flow rates and directions inside two models. In Fig 4, the inlet gas flow rate in the invasive model was half of that in the non-invasive model under the same inlet pressure. This means that the total resistance of the invasive model is approximately twice that of the non-invasive model. This result appears consistent with that of the clinical study by Berry et al. [43], which indicated that almost 40% of the total airway resistance is contributed by the ETT. As shown in Table 1B, the gas flow rates are highly dependent on the inlet pressure. This means higher ventilatory support can compensate for the resistance of the ETT. Till date, appropriate ventilatory support has been shown to have a lung protective effect [1, 2, 36, 44]. Therefore, we should recognize ETT as an external resistance [45–49] and should identify the resistive component from the total respiratory work [50–55]. Fig 4C represents the gas flow from the non-dependent toward the dependent lung under the solid-like effort condition, that is, the pendelluft. Moreover, more entrained flow from the upper and middle lobes was observed in the invasive model. This means that because of the resistance of the ETT, the gas flow from the inlet toward the dependent lung was limited, and instead, pendelluft from the non-dependent toward the dependent lung increased.

Fig 5 shows that in the invasive model, the jet flow from the ETT's tip moved toward the tracheal bifurcation and generated a relatively faster flow along the central axis in the tracheobronchial tree. Meanwhile, the faster flow generated at the glottis decelerated before it reached the bifurcation in the non-invasive model. With this, the energy distribution in the invasive model is more uneven than in the non-invasive model. As a result, as shown in Fig 4, the flow rate through each outlet was more homogenous in the non-invasive model than in the invasive model. Fig 6 shows entrained flow from the RU toward the tracheobronchial tree, even though the static pressure of the six outlet groups was set as 0 cmH$_2$O equally. This entrained flow can be explained by the venturi effect caused by the faster flow from the tip of the ETT. This phenomenon suggests that gas flow in the patient airway is not only determined by the inlet and outlet pressure gradient, but also affected by the ETT and patient airway geometry, and consequent uneven gas flow distribution.

## Comparative research 2: Effect of ventilatory support

Here, we discuss the effect of ventilatory support on pendelluft. Fig 7 indicates the gas flow into the model through some of the outlets (pendelluft) under the strong solid-like effort condition. In both the invasive and non-invasive models, greater inlet pressure decreased the pendelluft. This means that greater inlet pressure reduced the pendelluft during the early inspiratory phase. This is consistent with the results of the clinical study by Coppadoro et al. [7]. Specifically, in the non-invasive model, the increase in inlet pressure from a low to high support condition decreased pendelluft by 29%. Meanwhile, in the invasive model, pendelluft decreased by 11%. In addition, this suggests that clinically acceptable ventilation support does not completely prevent pendelluft during the early inspiratory phase. Fig 8 shows that in the

non-invasive model, higher inlet pressure increases gas flow out of the RU. Contrarily, in the invasive model, a higher inlet pressure decreases the gas flow out of RU. This may be related to the jet from the tip of the ETT tip. Higher inlet pressure resulted in greater jet velocity, leading to a more intense venturi effect. This can explain the limited effectiveness of increasing inlet pressure in improving pendelluft in the invasive model.

Additionally, we should discuss how this study is applied to the mechanical power (MP) and WOB. MP is an estimate of the mechanical energy per minute that is applied to the respiratory system. MP is divided into resistive, elastic, and PEEP components. In patients with ARDS, higher MP has been reported to be associated with higher mortality [56]. However, the individual impact of each component requires further discussion. For instance, some reports indicate that the resistive component may not be as damaging to the lung as initially expected [57, 58], elastic components above a certain value have been found to be particularly harmful [59], and the PEEP component has shown protective effects on the lungs under certain conditions [60]. In this model, the resistive component is expressed as the pressure difference between the inlet and outlet pressure, and elastic component is expressed as the outlet boundary conditions. Therefore, as inspiration proceeds and the pressure difference decreases (corresponding to the outlet boundary condition change from strong to weak), the inspiratory flow will decrease and less energy will be consumed as the resistive component. At the end of the inspiration, most of the energy is used to keep the lung inflated (elastic component). Regarding WOB, we should also consider the threshold load, which refers to the amount of work required to commence inspiratory flow; this is determined by auto-PEEP [61]. The inspiratory flow delay caused by the threshold load could worsen the pendelluft, because during this delay, the ventilatory support is zero, and more gas flow is assumed to be directed from the upper to the lower lobe.

## Limitations

This study had some limitations. First, the models did not take the peripheral airway into account, although the peripheral airway's contribution to total airway resistance has been reported as minimal [22]. Second, we simulated only steady flow under several boundary conditions. Clinically, the pressure and airflow in the segmental bronchi vary during the respiratory cycle [62]. Because it is difficult to accurately recreate the boundary conditions across the entire respiratory cycle, we examined three intensities of inspiratory effort in this study. In practical terms inspiration proceeds and the lungs are inflated, the outlet boundary condition changes (corresponding to the intensity changes from strong to weak). Furthermore, the pressure and gas flow are affected by many respiratory factors. For example, PEEP affects the lung compliance, strain distribution pattern [63], and intensity of the spontaneous breathing [64, 65]. Moreover, when higher chest wall compliance is assumed, a greater part of the patient's inspiratory effort is used to move the chest wall than for lung inflation. As the boundary conditions used in this study are integrated values of these factors, the influence of each factor is not considered separately. More clinical data, including data about the compliance of each lung lobe throughout the respiratory phases, will help us clarify the non-uniform inflation of the lung and regional strain [66]. Third, this study did not use individual tracheobronchial trees because the purpose of this study was to clarify the basic relationship between the ETT and tracheobronchial tree under various boundary conditions. Fourth, this study did not take the patients' position into account. Previous studies have reported that FRC in the supine position is smaller than that in the upright position; a smaller FRC is relevant to stronger diaphragm contraction, leading to more inter-lobar pressure difference caused by the solid-like behavior [1, 67]. Therefore, if we investigate the change in gas flow distribution due to the body

position, it may be appropriate to apply more negative pressure to the lower lobes in the supine position than in the upright position. Further research with the tracheobronchial tree based on computed tomography data is needed to resolve patient-specific ventilatory problems [68].

## Conclusions

Excessive negative pressure in the lower lobes generates inter-lobar pressure difference, causing pendelluft [11, 41]. This study shows that pendelluft caused by the inter-lobar pressure difference can be improved by increasing inlet pressure. However, this study also revealed that increasing inlet pressure enhances the jet flow from the ETT and causes a venturi effect, limiting the effectiveness of increasing inlet pressure. For this reason, increasing the ventilatory support would not prevent pendelluft completely during the early inspiration phase if the inspiratory effort is strong. At this point, reducing inspiratory effort through various means, including the use of muscle relaxants in patients with severe ARDS, is considered to be beneficial [69–71].

This study also showed that flow velocity and vector are not uniform on cross-section. Therefore, the jet from the ETT creates gas flow deflection in the tracheobronchial tree. This idea is often forgotten because the resistor-capacitor circuit analogy—often used to simplify respiratory physiology—does not deal with the three-dimensional shape of the airway. Further research on the morphological features of the ETT may decrease deflection and the venturi effect, consequently improving pendelluft.

We hope that future clinical studies will build on these simulations and lead to further research. These findings may also be applied clinically to optimize mechanical ventilation settings, reducing the repeated collapse and re-inflation pattern of the lung caused by pendelluft, which is thought to be harmful.

## Acknowledgments

We would like to thank Editage (www.editage.com) for English language editing.

## Author Contributions

**Conceptualization:** Kazuhiro Takahashi, Hiroaki Toyama.

**Data curation:** Kazuhiro Takahashi, Jinyou Yang.

**Formal analysis:** Kazuhiro Takahashi, Jinyou Yang, Takuji Ishikawa.

**Investigation:** Kazuhiro Takahashi, Jinyou Yang.

**Methodology:** Kazuhiro Takahashi, Kenji Kikuchi, Takuji Ishikawa.

**Project administration:** Takuji Ishikawa, Masanori Yamauchi.

**Resources:** Kenji Kikuchi, Takuji Ishikawa.

**Software:** Kazuhiro Takahashi, Jinyou Yang, Takuji Ishikawa.

**Supervision:** Yutaka Ejima, Takuji Ishikawa, Masanori Yamauchi.

**Validation:** Kenji Kikuchi, Takuji Ishikawa.

**Visualization:** Kazuhiro Takahashi.

**Writing – original draft:** Kazuhiro Takahashi, Hiroaki Toyama.

**Writing – review & editing:** Yutaka Ejima, Takuji Ishikawa, Masanori Yamauchi.

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
