## [Decision Letter · Decision Letter 0]

18 May 2023

PONE-D-23-07903Endotracheal tube, by the venturi effect, reduces the efficacy of increasing inlet pressure in improving pendelluftPLOS ONE

Dear Dr. Takahashi,

Thank you for submitting your manuscript to PLOS ONE. After careful consideration, we feel that it has merit but does not fully meet PLOS ONE’s publication criteria as it currently stands. Therefore, we invite you to submit a revised version of the manuscript that addresses the points raised during the review process. Please submit your revised manuscript by Jul 02 2023 11:59PM. If you will need more time than this to complete your revisions, please reply to this message or contact the journal office at plosone@plos.org. Please include the following items when submitting your revised manuscript:A rebuttal letter that responds to each point raised by the academic editor and reviewer(s). You should upload this letter as a separate file labeled 'Response to Reviewers'.A marked-up copy of your manuscript that highlights changes made to the original version. You should upload this as a separate file labeled 'Revised Manuscript with Track Changes'.An unmarked version of your revised paper without tracked changes. You should upload this as a separate file labeled 'Manuscript'.

We look forward to receiving your revised manuscript.

Kind regards,

Tai-Heng Chen, M.D.

Academic Editor

PLOS ONE

Reviewers' comments:

Reviewer's Responses to Questions

**Comments to the Author**

1. Is the manuscript technically sound, and do the data support the conclusions?

Reviewer #1: Partly

Reviewer #2: Yes

Reviewer #3: Partly

2. Has the statistical analysis been performed appropriately and rigorously? 

Reviewer #1: I Don't Know

Reviewer #2: N/A

Reviewer #3: Yes

3. Have the authors made all data underlying the findings in their manuscript fully available?

Reviewer #1: Yes

Reviewer #2: Yes

Reviewer #3: Yes

4. Is the manuscript presented in an intelligible fashion and written in standard English?

Reviewer #1: No

Reviewer #2: Yes

Reviewer #3: Yes

5. Review Comments to the Author

Reviewer #1: This bench research aimed to investigate via computer simulation whether endotracheal tube and ventilatory support contribute to pendelluft. Honestly, this paper is very hard to read and some sections are misleading.

I don't feel it suitable for publication in this journal.

Reviewer #2: 1. The reference to figure 1 has already been described in materials and methods (simulation model). So, there is no need to introduce it again in results (sentence 176).

2. Line 208. Sentence must be corrected: "and the gas flow into the models was not observed at the other outlets."

3. Figure 4: Even in the group with no effort, the outflow was greater in the LL and RL segments.

With the individual in the vertical position (standing or sitting), the base of the lung receives a greater volume of air than the apex. This model has been built in an upright position. How could the results be translated to a mechanically ventilated patient in the supine position? Put up for discussion.

4. When the airway resistance is increased the flow is totally turbulent, in this case there will be gas inertia and the linear velocity of the gas will be reduced. One concept that has emerged is the consideration of the amount of energy transferred from the ventilator to the respiratory system per unit time, which can be quantified as mechanical power (MP). Higher MP has been associated with worse clinical outcomes. It would be very interesting if the author could implement a MP calculation for the presented models. Put up for discussion.

Reviewer #3: Introduction

1) This model simulates regional biomechanical conditions that would occur in the context of increased respiratory effort in severe ARDS, at different frictional resistive components, with and without invasive MV support, apparently in supine position.

However, it is based only on works studying respiratory effort in subjects with ARDS under MV. There are some studies in recent years that have focused on both biomechanical phenomena (regional strain, lung heterogeneity, strain progression) and their regional biological transduction (lung damage progression) in murine and porcine models of P-SILI. It is relevant information to consider within the theoretical framework.

Method interesting and challenging. You use of finite element method to simulate regional conditions along bifurcations of the tracheobronchial tree. But there are certain methodological concerns that need to be clarified.

2) In “investigating the effect of an ETT on pendelluft”, I do not understand the rationale for using non-invasive models.

3) These model omits that the vigorous contraction of the diaphragm is relevant to regional strain generation in basal regions, resulting in an imperfect elastic anisotropic inflation and amplifying the regional lung injury (and regional progression of pulmonary edema, tidal recruitment, etc).

4) It also does not consider the use of PEEP, inspiratory time, flow morphology or the role of chest wall compliance.

5) As I understand it, it models resistive component (purely frictional) and a sum of elastic and viscoelastic component of the lung parenchyma (as a whole, without separating both), omitting threshold component.

6) It assumes that the flow will be lower in all the regions studied in invasive conditions than in non-invasive conditions, which in a clinical setting will depend on the ventilatory drive and the capacity to generate force by the respiratory muscles.

7) The authors should better explain how they defined (and measured) Pendelluft in their model.

8) Discussion

Pendelluft refers to regional displacement or redistribution of gas during the inspiratory phase. While it is a potential stress raiser in severe ARDS, I am not sure it is always harmful.

9) Part of the discussion and abstract should be redirected according to observations made.

6. PLOS authors have the option to publish the peer review history of their article (what does this mean?). If published, this will include your full peer review and any attached files.

Reviewer #1: No

Reviewer #2: **Yes: **Nazareth de Novaes Rocha

Reviewer #3: **Yes: **Pablo Cruces (https://orcid.org/0000-0001-9337-1254)

---

## [Author Response · Author response to Decision Letter 0]

11 Jul 2023

Point by Point response to each reviewer's comments has been uploaded seperately as "Response to Reviewers" file.

---

## [Decision Letter · Decision Letter 1]

29 Aug 2023

Endotracheal tube, by the venturi effect, reduces the efficacy of increasing inlet pressure in improving pendelluft

PONE-D-23-07903R1

Dear Dr. Takahashi,

We’re pleased to inform you that your manuscript has been judged scientifically suitable for publication and will be formally accepted for publication once it meets all outstanding technical requirements.

Kind regards,

Tai-Heng Chen, M.D.

Academic Editor

PLOS ONE

Reviewers' comments:

Reviewer's Responses to Questions

**Comments to the Author**

1. If the authors have adequately addressed your comments raised in a previous round of review and you feel that this manuscript is now acceptable for publication, you may indicate that here to bypass the “Comments to the Author” section, enter your conflict of interest statement in the “Confidential to Editor” section, and submit your "Accept" recommendation.

Reviewer #2: All comments have been addressed

Reviewer #3: All comments have been addressed

2. Is the manuscript technically sound, and do the data support the conclusions?

Reviewer #2: Yes

Reviewer #3: Yes

3. Has the statistical analysis been performed appropriately and rigorously? 

Reviewer #2: N/A

Reviewer #3: Yes

4. Have the authors made all data underlying the findings in their manuscript fully available?

Reviewer #2: Yes

Reviewer #3: Yes

5. Is the manuscript presented in an intelligible fashion and written in standard English?

Reviewer #2: Yes

Reviewer #3: Yes

6. Review Comments to the Author

Reviewer #2: The article opens the perspective, despite the limitations already described by the authors, to better understand the effects of Pendelluft through computing. In addition, it emphasizes the deleterious effects resulting from invasive ventilation.

Reviewer #3: congratulate the authors for the new version of their manuscript. Your biomechanical modeling allows a better understanding of regional respiratory mechanics, both in spontaneous and controlled ventilation.

7. PLOS authors have the option to publish the peer review history of their article (what does this mean?). If published, this will include your full peer review and any attached files.

Reviewer #2: **Yes: **Nazareth de Novaes Rocha

Reviewer #3: **Yes: **Pablo Cruces

---

## [Editor Report · Acceptance letter]

6 Sep 2023

PONE-D-23-07903R1 

Endotracheal tube, by the venturi effect, reduces the efficacy of increasing inlet pressure in improving pendelluft 

Dear Dr. Takahashi:

I'm pleased to inform you that your manuscript has been deemed suitable for publication in PLOS ONE. Congratulations! Your manuscript is now with our production department. 

Kind regards, 

on behalf of

Dr. Tai-Heng Chen 

Academic Editor

PLOS ONE